# Effects of Model Grid Spacing for Warm Conveyor Belt (WCB) Moisture Transport into the Upper Troposphere and Lower Stratosphere (UTLS)—

## **Part I: Lagrangian Perspective**

Cornelis Schwenk <sup>1</sup> and Annette Miltenberger <sup>1</sup>

<sup>1</sup>Institute for Atmospheric Physics, Johannes Gutenberg University Mainz

Correspondence: Cornelis Schwenk (c.schwenk@uni-mainz.de)

Abstract. Warm conveyor belts (WCBs) in extratropical cyclones transport moisture and hydrometeors into the upper troposphere and lower stratosphere (UTLS), influencing the radiative balance. Earlier research has shown that the horizontal grid spacing of numerical weather prediction (NWP) models has an impact on the modelled WCB properties, such as ascent rates and diabatic heating. This first part of a two-part study examines the impact of model grid spacing on the transport of moisture from a Lagrangian perspective. We analyze two ICON model simulations of one North Atlantic WCB case study: a convection-parameterizing run at ~13 km and a convection-permitting run at ~3.5 km approximate grid spacing. We hypothesize that key differences in the modelled transport of moisture arise from higher vertical velocities in the high-resolution simulation. The convection-permitting simulation produces more rapid ascent and a drier WCB outflow with lower specific and relative humidity. We attribute this to higher ice number concentrations, which deplete supersaturation more efficiently. This high-resolution simulation also exhibits more pronounced frozen-phase microphysics, stronger frozen precipitation, notably different hydrometeor mass mixing ratios, number concentrations, and radii than the lower resolution simulation — indicating that horizontal resolution substantially influences modelled WCB cloud composition. These results demonstrate that weather and climate models using convection-parameterizing resolutions may systematically misrepresent WCB cloud properties and UTLS humidity, with potential consequences for accurately simulating Earth's radiative budget and upper-level flow.

#### 15 1 Introduction

Water vapor is the most dominant greenhouse gas (Schneider et al., 2010), and in the upper troposphere / lower stratosphere (UTLS), variations in its concentration drive one of the most significant positive climate feedback mechanisms (Li et al., 2024; Held and Soden, 2000; Dessler et al., 2013, e.g.). Even minor changes in UTLS water vapor can substantially impact Earth's radiative budget (Wang et al., 2001; Hansen et al., 1984) and influence both mean and regional circulation patterns (Charlesworth et al., 2023; Ploeger et al., 2024, e.g.). However, even though water vapor measurements in the UTLS have increased over the past 20 years (Zahn et al., 2014; Jeffery et al., 2022; Hurst et al., 2011; Tilmes et al., 2010; Konjari et al., 2024), it remains one of the most poorly characterized regions of the atmosphere, mainly due to challenging measurement conditions (Jeffery et al., 2022). Nevertheless, in recent years multiple studies have tried to address this gap and to quantify the moisture content of the UTLS and associated transport pathways.

25

Several modelling studies (Ueyama et al., 2018, 2023; Dauhut et al., 2018; Hassim and Lane, 2010) and experimental studies (Corti et al., 2008; Danielsen, 1993; Lee et al., 2019; Gordon et al., 2024) have shown that deep convection in the tropics and the sublimation of detrained ice crystals are a major source of UTLS moisture. Homeyer et al. (2024) and Homeyer et al. (2014) specifically addressed the UTLS over the continental United States and confirmed that in this (limited) sample of the extratropics, overshooting convection remains an important contributor to UTLS moisture. Future warming of the tropopause is expected to increase UTLS moisture levels (Gettelman et al., 2010), while enhanced sublimation of ice stemming from convective activity is also projected (Dessler et al., 2016). Historically, transport directly into the tropopause region has increased from 2011 to 2020 compared to the 1980s (Jeske and Tost, 2025), but it is unclear how this will change in the future. Convection is however not the only transport pathway for UTLS moisture.

35

In addition to convection, warm conveyor belts (WCBs) were also identified as a major source of moisture in the extratropical UTLS (Zahn et al., 2014; Guo and Miltenberger, 2025). WCBs are large-scale ascending air streams that develop in extratropical cyclones (ETCs). They produce the elongated cloud bands typically observed in these systems, and over the course of roughly two days, WCBs transport moist boundary-layer air poleward and upward into the upper troposphere. During this time, the ascending air parcels experience complex microphysical and dynamical processes that lead to the formation of clouds and precipitation, affect upper-level wave propagation, and introduce both water vapor and hydrometeors into the UTLS (Madonna et al., 2014). Consequently, WCBs shape mid-latitude weather patterns, guide storm evolution, and influence Earth's radiative balance (Madonna et al., 2014; Rodwell et al., 2018; Joos, 2019). The large amounts of precipitation formed by WCBs often create hazardous conditions (Pfahl et al., 2014), with WCBs estimated to account for over 80% of total precipitation in Northern Hemispheric storm tracks (Pfahl et al., 2014; Eckhardt et al., 2004). Therefore, for weather forecasts it is important that models accurately represent WCBs, since their incorrect representation has been identified as a key factor in increasing forecast uncertainty (Pickl et al., 2023; Berman and Torn, 2019; Rodwell et al., 2018; Grams et al., 2018).

75

However, correctly representing WCBs is also important for climate forecasts (Joos, 2019). In the early stages of their lifetime, when WCBs are positioned in the southern extra-tropics, the ascending air produces low-level clouds that have a predominantly cooling effect. As the WCB moves northward and the air rises further, high-level frozen clouds have either a warming or a cooling effect, depending on solar radiation and cirrus optical thickness (Krämer et al., 2020; Joos, 2019). Mixed-phase clouds, present in the transition phase, contribute an uncertain net radiative effect. In the outflow region, WCBs are often associated with ice-supersaturated air masses (Spichtinger et al., 2005) and a cirrus cloud shield extending up to 3 km in depth (Binder et al., 2020), which usually results in a warming effect. A recent climatological study from Guo and Miltenberger (2025), which analysed ERA5 reanalysis data from 2010 to 2019, determined that, in December, January, and February, grid-scale advection accounts for up to 13.8% of the total moisture transport by WCB into the upper troposphere in the main ascent regions over the North Atlantic and Pacific ocean basins. However, it remains difficult to precisely quantify the amount of water vapor WCBs transport into the UTLS — and how this amount will evolve in a warming climate. As a result, the overall radiative impact of WCBs remains difficult to quantify, and how the balance between cooling and warming effects may shift under future climate conditions remains an open question (Joos, 2019).

Several factors make the accurate modelling of WCBs challenging. As ascending air undergoes warm-phase, mixed-phase, and cold-phase microphysical processes (Binder et al., 2020; Forbes and Clark, 2003; Gehring et al., 2020), the resulting clouds span a wide range of temperatures and pressures and exist in different phases. This variability adds significant complexity to their representation in models (Oertel et al., 2023; Schwenk and Miltenberger, 2024; Hieronymus et al., 2025). Therefore, it is unsurprising that the parameterisation of the microphysical processes in WCBs are a major source of numerical weather prediction (NWP) model uncertainty (Morrison et al., 2020; van Lier-Walqui et al., 2012; Posselt and Vukicevic, 2010). For instance, sensitivity experiments have shown that the choice of microphysical parameterisation schemes significantly influences WCB ascent characteristics (Mazoyer et al., 2021, 2023). Additionally, the diabatic processes occurring during WCB ascent are highly sensitive to the specific cloud microphysical parameters used within a given scheme (Hieronymus et al., 2022; Neuhauser et al., 2023; Forbes and Clark, 2003; Oertel et al., 2025). Schwenk et al. (2025) also show that changes in microphysical parameter choices in NWP models (such as the capacitance of ice and snow) within physically plausible ranges significantly alter the microphysical composition and ice content in the outflow of a WCB. These studies demonstrate the complexity of microphysical processes in WCBs and the substantial challenges they pose for accurate model representation.

Another factor that adds complexity and has recently gained attention is the role of embedded convection within WCBs (Rasp et al., 2016; Oertel et al., 2020; Schwenk and Miltenberger, 2024). Oertel et al. (2020) found that convective regions in WCBs differ from classical synoptic-scale slantwise ascending regions by producing enhanced surface precipitation, more intense cloud diabatic heating, and distinct imprints on the PV structure of the WCB outflow. Additionally, they observed that convective air parcels do not ascend continuously but experience intermittent periods of strong ascent and even descent. This finding was supported by Schwenk and Miltenberger (2024), who further showed that convective air parcels transport substantially more ice into the UTLS than their slower-ascending counterparts and experience different microphysical processes during the

100

105

110

ascent. They also identified differences in relative humidity distributions between fast- and slow-ascending air parcels, despite vapor conditions being primarily controlled by temperature. These studies, based on high-resolution convection-resolving simulations, raise an important question: Can low-resolution, convection-parameterising simulations adequately represent WCBs, particularly their role in moisture transport into the UTLS?

A number of studies have found that increasing model resolution (especially when moving from a resolution that requires convective parameterisation to one that explicitly resolves some fraction of convective activity) changes and improves NWP model results. Senf et al. (2020) showed that refining the grid spacing up to 2.5 km significantly improved simulated top-of-atmosphere cloud-radiative effects over the North Atlantic. Vergara-Temprado et al. (2020) found that fine-resolution models (exploring ranges from 50 km down to 2.2 km) with deep convection treated explicitly led to notable improvements in precipitation patterns and the diurnal cycle in annual simulations of the European climate. Homeyer (2015) determined that when changing the vertical and horizontal resolutions within the convection permitting regime from 3 km to 0.33 km for simulations of extratropical convection, the depth of overshooting and cross-tropopause transport increased with horizontal and decreased with vertical model resolution.

Focusing on WCBs, Choudhary and Voigt (2022) found that as grid spacing decreased gradually from 80 km to 2.5 km, more WCB air parcels ascended faster and higher into the atmosphere. They also observed an increase in diabatic heating at all pressure levels, primarily driven by cloud-associated phase changes in water. These results suggest that the representation of microphysical processes during WCB ascent are resolution-dependent, potentially reflecting the influence of convective air parcels. This is supported by the fact that the widely used Tiedtke-Bechtold convection parameterization scheme only provides a basic representation of cloud microphysics (Tiedtke, 1989; Bechtold et al., 2008). Additionally, Schwenk and Miltenberger (2024) showed that convectively ascending WCB air parcels undergo distinct microphysical processes than slowly ascending ones.

In spite of these various studies, so far, no study has examined how WCB moisture transport changes when transitioning from convection-parameterising to convection-permitting resolutions — a gap this paper aims to address. Therefore, our research question is: How does increasing model resolution from convection-parameterising (13.6 km effective grid spacing) to convection-permitting (3.2 km effective grid spacing) affect the transport of moisture and hydrometeors into the UTLS by a WCB? Given the results from Schwenk and Miltenberger (2024), we hypothesize that many of the differences observed between the simulations are primarily driven by changes in vertical velocity associated with higher model resolution.

This paper primarily focuses on comparing the representation of WCB processes in convection-parametrized and convection-permitting models, and analysing the resulting differences along WCB trajectories. Note that in the Lagrangian perspective, the differences are necessarily only identified in air parcels undergoing WCB ascent based on grid-scale wind. Therefore, the impact of the convection parameterisation is only seen as an additional tendency in the Lagrangian budget if it affects WCB

parcels. To address this issue and focus more on the overall outflow properties, we present an Eulerian analysis of the same simulations in the second part of the paper. Despite the limitations of the Lagrangian perspective, we shall see that it offers important physical process-oriented insight into the differences between the simulations, which cannot be gained by Eulerian analysis only.

This paper is structured as follows: First, we will describe the WCB case, the model setup, the online trajectories, and the Lagrangian diagnostics performed. This part is similar to Schwenk and Miltenberger (2024), since this is a follow-up study. We then show the results of the Lagrangian analysis, where we compare the trajectories for the global and the nested simulation. Since the Lagrangian data does not represent all relevant variables and does not necessarily fill the entire WCB domain, potentially leading to biased results, an Eulerian analysis is undertaken in Part-II (Schwenk and Miltenberger, 2025) of this study. This study largely reinforces the findings from this paper with the addition of investigating differences in radiation.

## 130 2 Methods

135

In this study we examine two simulations of a WCB case study, run at convection permitting and convection parameterising resolutions. This work builds on the study by Schwenk and Miltenberger (2024), since we examine the same case study and in the case of the convection permitting simulation, the same simulation output. The model setup for the convection parameterising simulation is very similar. Therefore, we will provide only a brief overview of the WCB case, model setup, and Lagrangian data analysis, and direct the reader to Schwenk and Miltenberger (2024) for further details.

## 2.1 WCB Case, Model Setup and Online Trajectories

This study investigates the same WCB case as Schwenk and Miltenberger (2024). From the afternoon of 22 September 2017 to the end of 23 September 2017, a strong cyclone with a characteristic WCB band dominated the weather over the northern Atlantic. The initial low pressure system formed off the coast of Newfoundland (approximately 52°N and 45°W) at 04:00 UTC on 21 September 2017, and underwent explosive cyclogenesis to form an extratropical cyclone over the central northern Atlantic 24 hours later at 04:00 UTC on 22 September 2017. On the morning of 23 September 2017, the northern edge of the cyclone reached Iceland and the eastern edge Ireland. By the evening it dissipated over northern Europe (for figures see Schwenk and Miltenberger (2024)).

We simulated this event using the Icosahedral Nonhydrostatic (ICON) model (v2.6.2; Zängl et al. (2014)). Two simulations were initialized with the operational ICON global analysis at 00:00 UTC on 20 September 2017, running for 96 hours until 00:00 UTC on 24 September 2017, when the WCB dissipated over northern Europe. The first simulation, from now on referred to as the "nested" simulation, has a setup which includes a global domain with two nested high-resolution grids, covering the main WCB ascent region (Fig. 1 b, Fig. A4 a). The global domain uses a R03B07 grid (~13 km effective grid spacing), while the nested domains employ R03B08 (~6.5 km) and R03B09 (~3.3 km) grids and are linked to each other and the global

160

domain by two-way coupling (Zängl et al., 2022). In the higher-resolution domains only shallow convection is parametrized, whereas in the global domain also deep convection is parametrized by the Tiedtke–Bechtold scheme (Tiedtke, 1989; Bechtold et al., 2008). Other sub-grid processes, including turbulence, orographic drag, and radiation, follow standard ICON schemes. Cloud microphysics are simulated with the two-moment scheme by Seifert and Beheng (2005), representing six hydrometeor species (cloud droplets, rain, ice, snow, graupel, and hail). The second simulation, from now on referred to as the "global" simulation, has the same model set-up and only differs in that it only has one global domain on a R03B07 grid (~13 km resolution; Fig. 1 a) and therefore parameterizes deep convection using the Tiedtke–Bechtold scheme everywhere.

To calculate the Lagrangian data we used the online trajectory module in ICON, developed by Miltenberger et al. (2020) and extended by Oertel et al. (2023) for domain nesting. In both simulations, trajectories were initiated every 5 hours throughout the 96-hour simulation, with output recorded every 30 minutes. Starting positions were randomly selected within a designated region across six vertical levels ( $\sim$ 1000–800 hPa). This region was determined using prior offline ERA5-based trajectories to ensure robust WCB representation (Fig. 1).

**Figure 1.** Simulation setup as well as pressure, latitude and longitude course of selected WCB trajectories for every fifth trajectory in the global simulation in a), and every tenth trajectory in the nested simulation in b). The online trajectory starting area is marked by the dashed lines, nested domain 2 (red) and 3 (black) boundaries are indicated in b) by the solid lines.

#### 2.2 WCB trajectory selection, ascent timescales, time-integrated process rates

Our selection algorithm identifies WCB trajectories based on two criteria. First, a WCB trajectory must ascend at least 600 hPa within 48 hours (a common definition of WCB ascent, see Madonna et al. (2014); Oertel et al. (2023)), and second, the trajectories must lie within two specified longitude-latitude regions at two distinct times, determined using Eulerian cloud cover and sea-level pressure data from our simulation. This ensured the trajectories were part of the WCB and not from unrelated mesoscale convective systems (MCSs) or other cyclones. This selection algorithm resulted in 113 993 WCB trajectories (from

180

in  $15.5 \cdot 10^6$  trajectories overall) for the global simulation, and 393 070 (from in  $91.8 \cdot 10^6$  trajectories overall) in the nested simulation.

For the characterization of trajectory ascent time we use the common ascent timescale  $\tau_{600}$  ( $\tau_{300}$ ) defined as the fastest time in which a trajectory ascends 600 (300) hPa (Rasp et al., 2016; Oertel et al., 2023; Schwenk and Miltenberger, 2024). To define the beginning and end of the ascent, we use  $\tau_{WCB}$ , which is defined by extending the  $\tau_{600}$ -period backwards and forwards as long as the ascent velocity remains above 8 hPa h<sup>-1</sup> (Schwenk and Miltenberger, 2024; Guo and Miltenberger, 2025). During  $\tau_{WCB}$ , a trajectory is defined as ascending within the WCB, before (after) this time it is defined as being part of the WCB inflow (outflow). To quantify the convection experienced per trajectory (during the ascent), we use the maximum pressure difference experienced by the trajectory in any 2 h time span ( $\max(\Delta p_{2h})$ ).

As in Schwenk and Miltenberger (2024), microphysical-process rates (such as deposition or riming) are set to zero at the start of  $\tau_{\rm WCB}$  and accumulated until the end of the ascent. The instantaneous process rates from ICON have the units  ${\rm kg\,kg^{-1}s^{-1}}$ ; we additionally have "integrated" process rates with the units  ${\rm kg\,kg^{-1}}$  along air mass trajectories (see Schwenk and Miltenberger (2024) for details).

## 2.3 Precipitation Efficiency (PE), Condensation Ratio (CR)

Two useful metrics to quantify the removal of moisture from an air parcel are the condensation ratio (CR) and the precipitation efficiency (PE) (Barstad et al., 2007; Miltenberger, 2014; Dacre et al., 2023). To remove water vapor by microphysical processes (as opposed to turbulence), it must be converted to hydrometeors that precipitate. The efficiency with which the conversion takes place is quantified by CR, and the efficiency with which the hydrometeors precipitate by PE. In this paper, we use the Lagrangian definitions of CR and PE from Schwenk and Miltenberger (2024) that take into account all relevant microphysical processes, as well as moisture lost or gained by the turbulence parameterisation, the convection parameterisation and numerical residuals. In formulation by Schwenk and Miltenberger (2024), CR represents the fraction of water vapor that is present at the start of the ascent, or is brought in by turbulence, convection, or numerical residuals, and that is converted into hydrometeors by the end of the ascent. PE measures the fraction of hydrometeors that either formed during the ascent, were present at the start, or were carried in by turbulence, convection, or numerical residuals, and that are precipitated by the end of the ascent. Both PE and CR are bounded by 0 and 1. For the full definitions and derivations see Schwenk and Miltenberger (2024).

## 3 Differences in properties of WCB trajectories from convection-permitting and -parameterising simulations

In this section, we use the online trajectory data to analyse and compare the ascent characteristics and moisture transport into 200 the UTLS by the WCB in both the global and nested simulations. Since this study is a follow-up to Schwenk and Miltenberger







(2024), the results for the convection-permitting simulation and their physical interpretation are largely identical to Schwenk and Miltenberger (2024).

#### 3.1 Ascent characteristics and evolution over the WCB lifetime

When comparing the ascent timescales  $\tau_{600}$  and  $\tau_{300}$ , the nested and the global simulation are very different (Fig. 2). The distributions of  $\tau_{600}$ -values are of a similar shape in both simulations (Fig. 2 a), with a single peak, a tail to larger values, and a sharp decline for smaller values. However, in the global simulation, peak is more flat, and the distribution is shifted to larger values, indicating a slower ascent overall. This is reflected in the mean (median)  $\tau_{600}$ , which is 17.0 h (14.5 h) and 21.4 h (20.0 h) for the nested and the global simulation, respectively. Because of this shift, the minimum  $\tau_{600}$ -value in either distribution is 4.5 h for the global simulation, compared to 0.5 h for the nested simulation. This means that the global simulation has virtually no convective trajectories when using the definition from Schwenk and Miltenberger (2024), which considers a trajectory to be dominated by convective motion if  $\tau_{600} 

trajectories (Figs. A3, A4 and A5) which each exhibit different large-scale ascending behavior. During TF1 (from 12:00 UTC 2017-09-20 to 10:00 UTC 2017-09-21; Fig. A3) trajectories are mainly located in the southern north Atlantic, are close together and are just beginning the large-scale ascent. However, most of this initial ascent is located outside of domain 3, where we want to focus our analysis (differences between the convection-parameterising and -permitting simulation will be most prominent in domain 3). In TF1, the mean (median)  $\Delta p_{2h}$  for both simulations does not exceed -10 hPa (-1 hPa).





In TF2 (00:00 UTC 2017-09-22 to 06:00 UTC 2017-09-23; Fig. A4) we see the strongest large-scale ascent that mostly occurs within domain 3, which is why we will focus most of the subsequent analysis on TF2. The mean (median)  $\Delta p_{2h}$  reaches values of approximately -32 hPa (-10 hPa) for the nested and -22 hPa (-12 hPa) for the global simulation. During TF2, the ascent is more vigorous in the nested simulation, with the largest 95<sup>th</sup> percentiles for  $\Delta p_{2h}$  of -150 hPa being more than 50 hPa larger (i.e. more negative) in the nested simulation than in the global simulation (Fig. 3 a). Histograms of pressure at the beginning and end of TF2 indicated that most trajectories complete their ascent within TF2 (Fig. 3 b). TF3 (10:00 UTC 2017-09-23 to 20:00 UTC 2017-09-23; Fig. A5) notes the time towards the end of the WCB lifetime, where most of the trajectories have completed their ascent, reached the UTLS, spread latitudinally and often propagated to regions outside of domain 3 (Fig. A5 a). TF3 is interesting because by this time most trajectories are part of the WCB outflow and they mostly remain in the UTLS at pressures below 400 hPa (Fig.A5 b). TF3 therefore allows us to investigate the outflow properties of the WCB.

Considering diagnostics separately for the three TFs allow us to focus on distinct areas of interest: i) the southern early stages of the WCB, where low- to mid-level liquid clouds dominate (not a focus in this study); ii) the period of strongest ascent, where a full range of mixed-phase clouds exists, and where we expect significant differences in microphysical processes and vertical hydrometeor distribution due to variations resolved ascent behaviour and the representation of the unresolved processes in the convection parameterisation (main focus in this study); and iii) the late stage, when most of the WCB trajectories have reached and remain in the UTLS as part of the outflow, allowing us to assess whether there are differences in hydrometeor and vapor content that remain even after the trajectories have left the highest-resolution domain.



Although our primary focus is on the ascent of WCB trajectories, we briefly point out interesting aspects of the descending motion of trajectories. In the nested simulation, Schwenk and Miltenberger (2024) found that most trajectories also experience periods of descent, a finding that is consistent with our analysis of the global simulation. However, the descending motion in the global simulation occurs at a different time and in a different location compared to the nested simulation. In the nested simulation, the strongest descents start from approximately 600 hPa and are spread out from 21.09.2017 and 23.09.2017 (Fig. A6 b). In contrast, the global simulation shows a bundled and concentrated period of descent starting at approximately 350 hPa and 12:00 on 22.09.2017 (Fig. A6 a). Some few trajectories in the nested simulation also experience intense descent of  $\Delta p_{2h} > 400$  hPa, whereas in the global simulation the maximum is 107 hPa. This indicates that not only the ascending but also the descending flow in the atmosphere changes when changing from a convection-parameterising to a convection-permitting resolution.

Figure 2. Histograms of  $\tau_{600}$  (a) and  $\tau_{300}$  (b) for the WCB trajectories in the global simulation (red) and nested simulation (black).

Figure 3. a)  $\Delta p_{2h}$ -values per trajectory over simulation run-time for the global (red) and the nested (black) simulation. Mean (solid line), median (dashed line), inter-quartile ranges (dark shaded) and 5<sup>th</sup> to 95th percentile ranges (light shaded) are plotted for the nested (black) and global (red) simulation. Vertical lines indicate the three time-frames (TF) that exhibit different ascent-behavior. b) shows histograms for the pressure of trajectories in the nested and global simulation at the beginning (dashed histograms) and end (solid-line histograms) of TF2.


#### 270 3.2 Conditions at the end of the ascent

In this section we summarise the properties of trajectories at the end of their WCB ascent (after  $\tau_{WCB}$ ) in the nested and the global simulation, and discuss the causes thereof.

## **Pressure, Temperature**

In both simulations, trajectories end their ascent at pressures between 200 hPa and 400 hPa (Fig. 4a), and at temperatures between -65°C and -20°C (Fig. 4b). The pressures at the end of the ascent are very similar (Fig. 4a), but at the end of the whole simulation, trajectories in the nested simulation are located on average at 319 hPa, compared to 328 hPa for the global simulation, indicating that they remain in higher regions of the atmosphere for longer. The average end-of-ascent temperature is -44.6°C in the nested and -44.0°C in the global simulation, and at the end of the simulation this differences grows to -41.4°C and -39.6°C (Fig. 4b).

At the end of the simulation, trajectories in the nested simulation at one average 24 h old (age defined as hours passed since end of ascent), compared to 25 h for the global simulation, which could in part explain the difference in pressure and temperature at this time. Because of the small shifts in the temperature and pressure distributions we will often compare variables between simulations at similar temperatures and pressures to eliminate biases (besides allowing for a more physics oriented analysis). Finally, the geographical distribution of trajectories at the beginning and end of the ascent, as well as at the end of the simulation, is similar in both simulations. The only notable difference is that in the nested simulation, WCB trajectories are found slightly further north and south at the end of their ascent (Fig. A1).

**Figure 4.** Normalized histograms of pressure (a) and temperature (b) at the end of the ascent (solid line histogram) as well as at the end of the simulations (dotted line histograms) for the nested (black) and global (red) simulations.



## Vapor content and relative humidity over ice

The logarithm of the specific humidity  $(\log_{10}(q_v))$  at the end of the ascent in the nested simulation follows a Gaussian distribution, whereas in the global simulation the distribution is skewed slightly towards larger values (Fig. 5 a). This results in a larger mean  $q_v$  value for the global simulation of 0.216 g/kg compared to 0.212 g/kg in the nested simulation. In both cases,  $q_v$  at the end of the ascent is primarily constrained by the temperature and pressure at the end of the ascent (Fig. A8), meaning that the higher  $q_v$  in the global simulation is primarily due to higher temperatures and lower pressures at the end of the ascent (Fig. 4 b). However, when looking at  $q_v$  values across pressure bins (Fig. 5 b), we find that between 225 hPa and 250 hPa, the mean, median and lower percentiles for  $q_v$  are consistently higher in the global simulation. This difference is also reflected when binning  $q_v$  by temperature (between -60 °C and -50 °C) but is harder to identify due to the much stronger dependence of  $q_v$  on T (and is therefore not shown).

We therefore also investigate the relative humidity over ice (RH<sub>i</sub>) at the end of the ascent. Here we find that, on average, RH<sub>i</sub> is slightly larger in the global simulation, with an average of 103.9 % compared to 103.4 % in the nested simulation (Fig. 5 c). This difference in mean values is pronounced when binned by pressure at approximately 250 hPa (Fig. 5 d). However, differences in the shapes of the RH<sub>i</sub> distributions are much more striking than the differences in mean values. In the nested simulation, RH<sub>i</sub> values have a sharp peak just below 100 % which is mostly absent in the global simulation (Fig. 5 c). In Schwenk and Miltenberger (2024), this peak was attributed to convectively ascending trajectories that have a higher condensation ratio and thus remove supersaturation more efficiently. Since convective trajectories are largely absent in the global simulation (Fig. 2), RH<sub>i</sub> assumes a more Gaussian distribution, with only a faint hint of a second peak below 100 %. These results clearly show that, at the end of the ascent and at pressures below 300 hPa, the vertical vapor distribution between the simulations differs greatly, and hint at the fact that this difference might be due to the lack of convectively ascending trajectories in the global simulation.

In order to determine a physical mechanism why some trajectories in the global simulation are more or less humid than their counterparts at the same pressure in the nested simulation when they finish their ascent, we adopt the same approach as Schwenk et al. (2025); Khvorostyanov (1995); Khvorostyanov and Sassen (1998); Khvorostyanov et al. (2001) and examine the relaxation timescale for supersaturation over ice ( $\tau_{\text{sat,ice}}$ ). For an air parcel with a vertical velocity of zero it is given by:

$$\tau_{\text{sat,ice}} = \frac{1}{4\pi \cdot \text{CAP} \cdot N_i \cdot r_i \cdot D_v} \quad \text{(with} \quad [N_i] = \text{m}^{-3}\text{)}, \tag{1}$$

with  $D_v$  the water vapor diffusion coefficient (taken as a constant value of  $3 \cdot 10^{-5} \,\mathrm{m}^2 \mathrm{s}^{-1}$ ) and CAP the capacitance of ice (0.5 for a spherical particle). This equation states that the time-scale for the removal of supersaturation is inversely proportional to the number concentration of ice  $(N_i)$  times the radius of ice  $(r_i)$ , and is only valid when vertical velocities are small (below  $\sim 0.1 \,\mathrm{m\,s^{-1}}$ ). The vertical velocities at the end of the ascent are  $(0.006 \pm 0.080) \,\mathrm{m\,s^{-1}}$  in the nested, and  $(0.010 \pm 0.032) \,\mathrm{m\,s^{-1}}$  in the global simulation. Therefore, we can safely use Equation 1 for our analysis when looking at the end of the ascent.







As in Schwenk et al. (2025), we find that  $\tau_{\rm sat,ice}$  plays a strong role in determining  $RH_i$  at the end of the ascent and that this dependence is the same in both the global and the nested simulation (Fig. A9). Because  $\tau_{\rm sat,ice}$ -values span many orders of magnitude, we look at the distribution of  $\log_{10}(\tau_{\rm sat,ice})$ -values at the end of the ascent and find that their shape is overall similar, but shifted to larger values for the global simulation (Fig. 5 e). The mean  $\log_{10}(\tau_{\rm sat,ice})$  in the global simulation is 3.56 ( $\sim 7000\,\mathrm{s}$  for  $\tau_{\rm sat,ice}$ ) compared to 3.44 ( $\sim 5000\,\mathrm{s}$  for  $\tau_{\rm sat,ice}$ ) in the nested simulation. When binned by pressure,  $\log_{10}(\tau_{\rm sat,ice})$  is larger in the global simulation from 225 hPa to 350 hPa corresponding to the pressure range where  $RH_i$  is also either larger or similar to the values in the nested simulation. Below 225 hPa,  $\log_{10}(\tau_{\rm sat,ice})$  in the global simulation is smaller, just as  $RH_i$ . Therefore,  $\tau_{\rm sat,ice}$  seems to be a suitable diagnostic to explain difference in  $RH_i$  at tend of the ascent and in particular suggest that difference in the hydrometeor population between both simulations are causing the variations in the  $RH_i$  distribution. Differences in the hydrometeor population between global and nested simulation are explored in the next paragraph.

## 335 Hydrometeor Content

In both simulations, almost all trajectories are fully glaciated by the end of the ascent, with the only remaining hydrometeor species being ice (not shown). However, the ice mass and number concentrations  $(q_i \text{ and } N_i)$  at the end of the ascent are very different between the simulations.  $q_i$  is on average 0.025 g/kg for the nested simulation, which is almost twice as high as the average of 0.015 g/kg for the global simulation (Fig. 6 b). The distributions for  $\log_{10}(q_i)$  assume a mostly Gaussian shape in both simulations, but are more narrow and shifted to smaller values for the global simulation (Fig. 6 c).  $N_i$  is on average 4 times higher in the nested simulation ( $\sim 4 \cdot 10^5$  1/kg) than in the global ( $\sim 10^5$  1/kg) (Fig. 6 a). Both  $\log_{10}(N_i)$ -distributions have a tail towards large  $N_i$  and are similarly shaped, but again the distribution from the global simulation is shifted to smaller values. The distribution for  $r_i$  is shifted to larger values in the global simulation (Fig. 6 e), and  $r_i$  values are clearly larger on average at pressures below 300 hPa (Fig. 6 f). This indicates that in the global simulation, the overall ice content at the end of the ascent is smaller in mass and number, while ice crystal radii are larger.

When binned by pressure,  $q_i$  and  $N_i$  at the end of the ascent are smaller in the global simulation in the same pressure regions (250-225 hPa) where  $q_v$  and  $RH_i$  are larger in the nested simulation or roughly the same both simulations (Fig. 6 b and d). Below 225 hPa, this behaviour is inverted, with  $q_i$  and  $N_i$  being larger in the global simulation. This inversion suggests that, when changing from a convection-parameterising to a convection-permitting resolution, the amount and vertical distribution of ice transported to the UTLS change. Above 300 hPa, the ice crystal radius  $(r_i)$  in the global simulation is similar to the nested simulation (Fig. 6 g), and below this pressure it is consistently larger in the global compared to the nested simulation (Fig. 6 f). Nevertheless, when averaged over all altitudes in the outflow,  $r_i$  is larger in the global simulation ( $\sim$ 0.11 mm on average) than in the nested simulation ( $\sim$ 0.10 mm on average), reflecting the higher  $N_i$  and  $q_i$  in the nested simulation. The distribution shapes of  $r_i$  are similar, with the slight difference that in the nested simulation, there is a hint of a secondary peak at very small radii (Fig. 6 e).

Figure 5. Normalized histograms of  $\log_{10}(q_v)$  (a),  $RH_i$  (d) and  $\log_{10}(\tau_{\rm sat,ice})$  (f) at the end of the WCB ascent. The corresponding distributions binned by pressure are shown in b) and c) for  $\log_{10}(q_v)$ , in e) for  $RH_i$  and g) for  $\log_{10}(\tau_{\rm sat,ice})$ . Mean (solid line), median (dashed line), inter-quartile ranges (dark shaded) and 5th to 95th percentile ranges (light shaded) are plotted for the nested (black) and global (red) simulation.

Figure 6. Normalized histograms of  $\log_{10}(N_i)$  (a),  $\log_{10}(q_i)$  (c) and  $r_i$  (e) at the end of the ascent. The distributions binned by pressure are shown in b) for  $\log_{10}(N_i)$ , in d) for  $\log_{10}(q_i)$  and f) for  $r_i$ . Mean (solid line), median (dashed line), inter-quartile ranges (dark shaded) and 5th to 95th percentile ranges (light shaded) are plotted for the nested (black) and global (red) simulation.

In the introduction, we hypothesized that the differences we see between simulations for variables such as ice crystal content and size at the end of the ascent will likely be due to larger vertical velocities in the nested simulation. We have already shown that overall, the ascent timescales are much smaller in the nested simulation (Fig. 2). Even for trajectories with similar  $\tau_{600}$ , the  $\max(\Delta p_{2\rm h})$ -values are much larger than in the nested simulation (Fig. A7 a). The intense convective motion for  $\tau_{600} 

390

395

Schwenk and Miltenberger (2024) determined that high vertical velocities allow for more ice crystals to remain suspended in an air parcel, and that the less time spent in the ascent means that there is also less time for the growth of larger ice crystals. Therefore, the differences in ascent velocity and time between the two simulations can plausibly explain the higher  $N_i$  and  $q_i$  for pressures larger than 225 hPa in the nested simulation, as well as the overall smaller  $r_i$ . However, we also found that for pressures below 225 hPa, this behaviour is inverted, with  $N_i$  and  $q_i$  being larger in the global simulation (differences in  $r_i$  are unchanged). This inversion cannot be explained by differences in ascent velocity. Instead, we hypothesize that the convection parameterization, which can transport ice to high altitudes in a single time-step, might lead to the higher  $N_i$  and  $q_i$  values for the global simulation above 225 hPa. This hypothesis is tested in the second part of this study.

The differences in vapor and hydrometeor content at the end of ascent that we discuss in this section suggest that trajectories in the two simulations undergo distinct (micro)physical processes during their ascent. This will be discussed in section 3.3.

These results also raise the question of whether the condensation ratio (CR) and precipitation efficiency (PE) are different in the nested and global simulations and is discussed next.

## **Precipitation Efficiency, Condensation Ratio**

We use the definitions of the Lagrangian precipitation efficiency PE and condensation ratio CR from Schwenk and Miltenberger (2024) to quantify the water budget of WCB air parcels, the efficiency of vertical moisture transport and the role of cloud processes in modifying the latter. CR quantifies the fraction of vapor present at the beginning the WCB ascent that is converted into hydrometeors by the end of the ascent. It is therefore a measure of how efficiently vapor is removed. PE quantifies the fraction of hydrometeors formed during the ascent that are lost by the end of the ascent. It is therefore a measure of how efficiently hydrometeors are removed.

Based on the overall lower  $q_v$  for the nested simulation discussed in the previous section we expect that the CR should be larger overall than in the global simulation, since a lower  $q_v$  generally indicates that more vapor has been converted and likely dominates over differences in the boundary layer water vapor field. The distribution of CR confirms this (Fig. 7 a). However, if considering CR for trajectories arriving at different altitudes in the outflow, CR is also larger in pressure bins where  $q_v$ -values are the same or similar (Fig. 7 c), indicating an overall greater fraction of vapor to condensate conversion in the nested simulation. This difference likely derives from differences in boundary-layer moisture field and representation of boundary-layer moisture transport in the two simulations: Trajectories in the nested simulation are more humid and warmer at the beginning of  $\tau_{\text{WCB}}$  (Fig. A2), and CR strongly depends on the initial  $q_v$ .

However, the higher overall  $q_v$  at the beginning of the ascent the nested simulation is most likely not due to a higher  $q_v$  in the planetary boundary layer (PBL) WCB inflow region of the nested simulation. Rather, the higher initial  $q_v$  is likely a result of the algorithm that determines the beginning of  $\tau_{WCB}$ : The algorithm goes backwards from the beginning of  $\tau_{600}$  until the ascent velocity is smaller than 8 hPa h<sup>-1</sup>, and since trajectories in the nested simulation generally experience higher vertical

420

velocities, this point can be shifted further backwards than in the global simulation. This higher overall ascent velocity in the nested simulation therefore shifts the beginning of  $\tau_{WCB}$  closer to the surface, where  $q_v$  is higher. We conclude that this is a potential weakness and source of error when calculating CR. Perhaps a future study should explore ways to account for this induced bias.

In the nested simulation, the PE has a more broad distribution than in the global simulation and is smaller on average (Fig. 7 b). The bias induced for CR from the  $\tau_{\text{WCB}}$  algorithm does not effect PE, because it depends more strongly on the initial hydrometeor concentration and not the vapor content. When binned by pressure, PE is smaller in the nested simulation from approximately 250 hPa to 350 hPa, and larger below 225 hPa (Fig. 7 d), which is mostly consistent with differences in  $N_i$  and  $q_i$  as expected (Fig. 6 b and d). This consistency confirms that the differences in hydrometeor transport, rather than the differences in vapor-to-hydrometeor conversion, drive the differences in hydrometeor content at the end of the ascent between the two simulations.

The larger spread in PE values for the nested simulation results in some trajectories precipitating nearly all hydrometeors, with 5% of trajectories in the nested simulation having PE > 99.999%, as opposed to just 0.5% in the global simulation. The retention of hydrometeors strongly depends on vertical velocities as well as the hydrometeor size and type. Since Schwenk and Miltenberger (2024) showed that convective activity changes which hydrometeor species are more abundant in an ascending air parcel, we are able to attribute the different spread of PE distributions to the more convective flow in the nested simulation.

In this section, we analysed the conditions of WCB trajectories at the end of their ascent. We found that trajectories in the nested simulation end their ascent at slightly lower pressures and temperatures than in the global simulation (Fig. 4). The vapor content is smaller overall in the nested simulation (Fig. 5 a), but especially so below pressures of 250 hPa (Fig. 5 b).  $RH_i$  is also smaller in the nested simulation and the distribution peaks below 100 %, whereas in the global simulation the distribution peaks above 100 % (Fig. 5 c and d). We are able to trace these differences to different relaxation timescales  $\tau_{\text{sat,ice}}$  between the two simulations.  $\tau_{\text{sat,ice}}$  is inversely proportional to  $N_i$  (Eq. 1), which at the end of the ascent is much higher in the nested simulation (Fig. 6 a and b), which explains why  $\tau_{\text{sat,ice}}$  is smaller and thus also  $RH_i$ . By analysing the moisture transport through the lens of the CR and PE, we are able to trace these differences in hydrometeor content (and therefore vapor) to differences in hydrometeor transport during the ascent (quantified by PE), as opposed to differences in the vapor-to-hydrometeor conversion (quantified by CR).

## 3.3 Microphysical composition and processes in the WCB ascent stage

In this section, we examine key differences and similarities in the microphysical processes and hydrometeor populations between the two simulations during TF2, the time period when the main WCB ascent occurs (Fig. A4). The aim is to develop a deeper understanding of what leads to the diagnosed differences in the outflow hydrometeor populations. In this analysis, it is important to note that the geographical distribution of trajectories is very similar between the simulations at each time step

(Fig. A1), allowing us to assume that any differences we observe arise from the simulated processes themselves rather than from geographical shifts between simulations.

Figure 7. Normalized histograms of  $\log_{10}(1-\mathrm{CR})$  in a) and of  $\log_{10}(1-\mathrm{PE})$ . The distributions binned by pressure are shown in c) and d), respectively, with mean (solid lines), median (dashed lines), inter-quartile ranges (dark shaded) and  $5^{\mathrm{th}}$  to  $95^{\mathrm{th}}$  percentile ranges (light shaded) for the nested (black) and global (red) simulation. We chose to display the parameters this way because it would otherwise be very difficult to visualize the difference in the CR and PE, which are both close to one and not spaced linearly. Larger negative values of  $\log_{10}(1-\mathrm{X})$  mean X is closer to 1. Therefore, values for CR and PE which are closer to one lie to the right in a) and b) (note the flipped x-axes), and to the top in c) and d) (note the flipped y-axes).

#### Microphysical processes

440

Firstly, we examine integrated process rates that are accumulated by the end of the ascent. In a previous study, Schwenk and Miltenberger (2024) showed that convective WCB trajectories exhibit higher accumulated frozen precipitation flux — defined as the net loss of frozen hydrometeors along the trajectory  $(q_{x,in} - q_{x,out})$ , integrated over  $\tau_{WCB}$  — while slow trajectories tend to produce more rain. We find that this pattern is similarly reflected in the global and nested simulations: Compared to the global simulation, the nested simulation shows a greater integrated frozen precipitation flux, a smaller integrated rain flux, and a higher total hydrometeor flux overall (Fig. 8,a). We also find higher integrated riming and condensation rates in the nested simulation, while the global simulation shows slightly higher integrated deposition rates (Fig. 8,b). These difference very likely

450

arise from the missing representation of large (convection-related) vertical velocities in the global simulation and the impact of ascent timescale on the cloud microphysical pathways as identified by Schwenk and Miltenberger (2024).

Moisture loss via the convection parametrisation is nearly zero in the nested simulation (Fig. 8,b), since most of the ascent occurs within domains 2 and 3 (Fig. A4 a), where the (deep) convection scheme is switched off. This lack of the convection parametrization for most of the ascent in the nested simulation can explain the large differences observed in integrated condensation (cond) and turbulence (turbs) rates between the simulations (Fig. 8,b): Without the option to redistribute moisture through the convection parametrisation, the higher resolution model must instead condense and precipitate water vapor via the cloud microphysics parametrisation or lose moisture to the turbulence parameterization.

The greater moisture loss due to turbulence in the nested simulation also supports our interpretation that the atmospheric flow in the nested simulation has more fine-scale structure increasing also wind shear and thus the loss of moisture due to turbulence. In the global simulation, approximately  $1.5\,\mathrm{g\,kg^{-1}}$  of moisture is lost due to the convection scheme, and moisture lost to the turbulence scheme is small. The clear signal from the convection parameterization demonstrates that the Lagrangian diagnostics are able to pick up on the contributions from the convection parameterization scheme, even though it does not influence the wind fields with which trajectories are advected. The negative sign also indicates that the contributions to the moisture budget mostly occur in the beginning of the ascent, where the scheme removes moisture from a parcel to redistribute it vertically, as opposed to at the end of the ascent, where the scheme takes moisture from the PBL and redistributes it into the parcel.

Binning the instantaneous (not integrated) microphysical process rates by pressure and temperature (Figs. A10 and A11) indicates that they differ primarily in magnitude rather than being shifted to different pressure or temperature regions. Notable exceptions are riming (Fig. A10 a, d) and the influx of graupel and rain (Fig. A11 a and b): in the global simulation, riming and the influx of graupel are largely absent below approximately 500 hPa and  $-10^{\circ}$ C, whereas in the nested simulation, trajectories still undergo substantial riming and experience graupel influx at pressures around 400 hPa and temperatures as low as  $-20^{\circ}$ C. This behaviour is due to the higher ascent velocities in the nested simulation, which allow for the production and retention of larger amounts of graupel and rain at lower pressures and temperatures.

Higher vertical velocities keep particles in suspension longer which allows them to grow. This means that the nested simulation favours the production of graupel over ice and snow (supported by the larger integrated riming rate shown in Fig. 8 b). Trajectories in the global simulation consistently experience a slightly greater influx of ice and snow than those in the nested simulation (Fig. A11 c and d). We note that these differences only slightly change the pressure and temperature of glaciation (first time step where liquid water content is zero; Fig. A12). On average, air parcels in the nested simulation glaciate at 362 hPa and -32.2 °C, compared to 356 hPa and -32.8 °C in the global simulation. Even though we attribute most of the previously identified differences between the two simulations to the difference in vertical velocities, the process of reaching full glaciation is not strongly affected by this difference. This is probably because vertical velocities between the two simulations differ most

strongly in the lower troposphere ( $p 

**Figure 8.** Box plots of a) the total hydrometeor flux, rain flux and frozen hydrometeor flux accumulated by the end of the ascent, as well as b) of the end-of-ascent accumulated microphysical process rates for riming, deposition, Wegener-Bergeron-Findeisen process, condensation, evaporation, turbulence parametrisation tendencies and convection parametrisation tendencies, respectively. All tendencies are shown separately for the nested (black) and global (red) simulation.

## **Hydrometeor content**

During the ascent, i.e. TF2, the total hydrometeor and frozen hydrometeor mass mixing ratio are both higher in the nested simulation across all pressure bins (Fig. 9 a and b). The mass mixing ratio of liquid water (Fig. 9 c) is similar at pressures of approximately 600 hPa but larger in the nested simulation above and below this pressure. When examining the number concentrations for individual hydrometeor species (Fig. A13 and Fig. A14), we find that the largest difference in frozen hydrometeor content is seen for graupel, which has far higher mass mixing ratios and number concentrations in the nested simulation (Fig. A13 c and f), especially below pressures of 600 hPa. The radii of ice and snow are smaller or similar in the nested simulation (Fig. A13 g and h), and the number concentrations mostly larger (Fig. A13 d and e). The mass mixing ratios are similar or slightly larger (Fig. A13 a and b). Especially for snow, the largest difference between simulations is seen for the size and





number concentration, not the mass mixing ratio. Cloud droplet mass mixing ratios and number concentrations are higher in the nested simulation (Fig. A14 a and c), whereas radii are similar. For raindrops, the mass mixing ratios and radii are larger in the nested simulation (Fig. A14 b and f) but the number concentrations are far lower (Fig. A14 d).

These differences can mostly be explained by the different vertical velocities experienced by trajectories in the nested and global simulation. Higher vertical velocities mean that more hydrometeors can remain suspended in the ascending air parcel, which explains higher mass and number mixing ratios in the nested simulation. The larger number and mass mixing ratios of cloud droplets (Fig. A14 a and c) could also be due to differences in CCN activation. This process is also strongly dependent on vertical velocity, with higher vertical velocities leading to higher CCN activation rates and smaller cloud droplets. Therefore, higher vertical velocities also impact the size distribution of cloud droplets, and as a consequence thereof also influences the conversion rates of cloud droplets to raindrops, and the size of the ice particles that result from the freezing of cloud droplets. Therefore, differences in vertical velocities can also explain the different hydrometeor radii. The larger raindrops seen in the nested simulation are exempt from this explanation, and could instead be caused by the higher influx of graupel from aloft (Fig. A11 b) which is melted to form raindrops.

**Figure 9.** Total (a), frozen (b) and liquid (c) hydrometeor mass mixing ratios binned by pressure for TF2. Mean (solid line), median (dashed line), inter-quartile ranges (dark shaded) and 5th to 95th percentile ranges (light shaded) are plotted for the nested (black) and global (red) simulation.

In summary, the trajectories in the two simulations undergo different microphysical processes during WCB ascent phase with arguably the most important difference being a shift of precipitation to the frozen phase in the nested simulation. During the ascent and for ice, snow, and cloud drops, the nested simulation tends to favour more and smaller hydrometeors, whereas for graupel and rain the opposite is the case. Differences in the vertical distribution of hydrometeors are minor for all hydrometeor




species except for graupel, which is shifted substantially to lower pressures in the nested simulation. We can plausibly attribute all of these differences to the different ascent velocities experienced by trajectories in the two simulations.

#### 3.4 Conditions after the end of ascent

In this section we investigate trajectories after they have completed their ascent, when they form the WCB outflow and remain mostly below pressures of above 500 hPa. This is the case in TF3, during which  $q_v$  and  $RH_i$  become overall very similar in the two simulations (Fig. 10 a, b). However, when binned by pressure, the mean and median  $q_v$  and  $RH_i$  are both higher in the global simulation than in the nested simulation below approximately 280 hPa (Fig. 10 a and b). In the case of  $q_v$ , this difference is relatively small in absolute values, but percentage-wise and at 250 hPa,  $q_v$  is 12% higher in the global simulation than in the nested simulation.

The vertical shift in ice-content we showed in Section 3.2 is even more pronounced in TF3.  $q_i$  and  $N_i$  are larger in the nested simulation below approximately 350 hPa, and larger in the global simulation above this pressure (Fig. 10 c and d). This is presumably due to the smaller  $r_i$  below 300 hPa at the end of the ascent in the nested simulation (Fig. 6 f), implying that the smaller ice can remain suspended for longer. This indicates that the nested simulation also produces more high-level cirrus clouds. The higher  $N_i$  in the nested simulation at the lowest pressures could also enhance vapor-to-ice conversion efficiency and therefore explain the smaller  $RH_i$ . However, this difference is minor and the larger  $RH_i$  in the global simulation below approximately 280 hPa might instead (or also) be caused by the influx of moisture from the convection scheme. This might also explain why  $q_i$  in the global simulation is slightly larger at approximately 250 hPa and  $N_i$  increases in the same region.

These differences in vertical distribution of  $q_i$ ,  $N_i$  and  $q_v$  in the outflow of the WCB between the simulations raise an important follow-up question: do these differences translate into detectable changes in the radiative balance at the top of the atmosphere (TOA)? While different values of  $q_i$  and  $N_i$  in the upper parts of the cloud can alter shortwave radiative fluxes and outgoing longwave radiation (via shifts in effective cloud top temperature), altered values of  $q_v$  in the upper troposphere above the cloud tops may alter water vapor greenhouse effect locally. As the investigation of radiative budgets from a Lagrangian perspective is not straightforward, we will only investigated these questions in more detail in Schwenk and Miltenberger (2025), where we present an Eulerian analysis of the WCB case.

## 4 Conclusions and Discussion

In this paper, we study the effects of different grid spacings on the modelled transport of moisture by WCBs into the UTLS. To this end, we consider a WCB case study that occurred on 23 September 2017 and investigate two ICON simulations run at grid spacings of  $\sim 13\,\mathrm{km}$  (convection parametrizing) and  $\sim 3.5\,\mathrm{km}$  (convection permitting). In this paper—which is part one of a two-part study—we analyse only the Lagrangian trajectory data. In Schwenk and Miltenberger (2025) we analyse Eulerian data to verify the results from this study and additionally investigate differences in radiation. While previous studies such


Figure 10.  $q_v$  (a),  $RH_i$  (b),  $q_i$  (c) and  $N_i$  (d) binned by pressure for TF3 (time after the ascent), with mean (solid lines), median (dashed lines), inter-quartile ranges (dark shaded) and  $5^{\text{th}}$  to  $95^{\text{th}}$  percentile ranges (light shaded) for the nested (black) and global (red) simulation.

as Choudhary and Voigt (2022) have investigated how grid spacing influences WCB trajectories, our focus on the transport of moisture and hydrometeors into the UTLS is novel. Additionally, we use online trajectories which deliver more detailed insight into cloud microphysical processes and convective ascent compared to offline trajectories (Miltenberger, 2014, e.g.).

We expect that WCB trajectories in the higher-resolution ("nested") simulation ascend more rapidly and that the observed differences in moisture and hydrometeor transport between the simulations can largely be traced back to this difference in ascent dynamics. Choudhary and Voigt (2022) also found that higher resolutions lead to higher vertical velocities, which is expected because in the lower-resolution ("global") simulation convection is parameterized using the Tiedtke–Bechtold scheme and hence represents subgrid-scale convective mass fluxes but does not explicitly generate resolved vertical winds. By contrast, in the convection-permitting nested simulation, CAPE is depleted through explicitly resolved ascending motions, leading to stronger vertical velocities. We put forward the hypothesis that this difference in vertical velocities affects moisture and hydrometeor transport, because Schwenk and Miltenberger (2024) showed that trajectories undergoing rapid ascent differ sub-







stantially in these properties compared to those that ascend more slowly, and because the hydrometeor and moisture transport in the convection parameterization is calculated differently to the convection permitting simulation.

In line with our expectations, trajectories in the nested simulation ascend more rapidly. Not only are the  $\tau_{600}$  and  $\tau_{300}$  values on average much smaller (Fig. 2), but the  $\max(\Delta p_{2h})$  values are also much higher even when  $\tau_{600}$  are similar (Fig. A7 a), indicating that trajectories in the nested simulation experience more frequent and intense "bursts" of convective ascent than in the global simulation regardless of overall WCB ascent time. Trajectories in the nested simulation also end their ascent at slightly lower pressures and temperatures than those in the global simulation (Fig. 4), consistent with Choudhary and Voigt (2022). We additionally find that trajectories in the nested simulation tend to remain at lower pressures and temperatures for longer. These results show that the atmosphere (in the WCB region) of the nested simulation has stronger vertical motions compared to the global simulation. In this study, we are able to plausibly trace many of the differences in hydrometeor and moisture conditions detected between simulations to these difference in vertical velocities.

At the end of the ascent, trajectories in the nested simulation have slightly smaller  $q_v$  and  $RH_i$  than in the global simulation (Fig. 5 a-e), but this does not hold true across all pressure bins.  $q_v$  and  $RH_i$  are smaller in the nested simulation for pressures between approximately 225 hPa to 300 hPa, and higher below 225 hPa. This indicates that when changing from a convection parameterizing to a convection permitting simulation, the vertical humidity distribution in the WCB outflow changes. We are able explain these differences using the relaxation timescale for supersaturation over ice ( $\tau_{\rm sat,ice}$ , Eq. 1), which is inversely proportional to  $N_i$  and  $r_i$ .  $N_i$  and  $q_i$  are higher in the nested simulation when  $RH_i$  is lower, and vice versa (Fig. 6 a-d), which makes differences thereof the likely explanation for the differences in  $RH_i$  and  $q_v$ . We attribute higher  $N_i$  and  $q_i$  in the nested simulation to higher ascent velocities that allow for the retention of more hydrometeors. These conclusions are consistent with the findings from Schwenk and Miltenberger (2024). The higher  $N_i$  and  $q_i$  in the global simulation, however, are attributed to different transport of hydrometeors by the convection scheme, not to slower ascent velocities. Finally, we find that  $r_i$  is generally smaller in the nested simulation (Fig. 6 e-f). However, this does not influence  $\tau_{\rm sat,ice}$ , because the differences in  $N_i$ , which span many orders of magnitude, dominates the denominator in Eq. 1.

During the ascent (TF2), microphysical processes in the nested simulation are more dominated by frozen processes such as riming and trajectories also produce more frozen precipitation compared to the global simulation (Fig. 8, Fig. A10). Hydrometeor content and radii during the ascent are also altered; in the nested simulation there is higher graupel mass and number concentrations and raindrops are larger. For ice, snow, and cloud drops, the nested simulation tends to favor more and smaller hydrometeors (Fig. A13, Fig. A14). Assuming that differences in vertical velocity are the cause of these differences, these results align with those of Schwenk and Miltenberger (2024) regarding the impact of vertical velocity variations within a single WCB case, suggesting that differences in vertical velocity are a plausible, physically-consistent mechanism for these changes.

After the end of the ascent, when trajectories are in the outflow of the WCB (TF3), differences in  $q_v$  and  $RH_i$  between

simulate Earth's radiative budget or the upper level flow.




the two simulations are diminishing. However, in the global simulation  $q_v$  remains slightly larger overall and  $RH_i$  is larger below approximately 280 hPa (Fig. 10 a and b). This indicates that the differences found at the end of the ascent can persist for some time and influence the WCB outflow on a longer timescale. On the other hand, the differences in  $q_i$  and  $N_i$  in TF3 are much more pronounced than at the end of the ascent, with both  $q_i$  and  $N_i$  being much larger in the global simulations for pressures larger than approximately 350 hPa and much smaller below (Fig. 10 c and d). However, at approximately 250 hPa in the global simulation,  $q_i$  is slightly larger than in the nested simulation, and  $N_i$  increases slightly, which might be caused by the transport of hydrometeors by the convection parameterization. Whether these differences in UTLS humidity and hydrometeor content translate to differences in other properties of the WCB outflow, such as radiative properties, is discussed in Schwenk and Miltenberger (2025).

The results of this study support our hypothesis that higher vertical velocities in convection-permitting simulations not only alter the ascent characteristics but also have a substantial effect on the transport of moisture and hydrometeors into the UTLS by a WCB, as well as the cloud composition and microphysical processes during the ascent. These findings are important because they suggest that convection-parameterizing simulations, which remain widely used in weather forecasts and especially in climate models, may inaccurately represent WCB cloud properties and humidity in the outflow (if we assume that higher-resolution models represent reality more accurately). This has potentially important implications for how these models

There are some important limitations to consider when interpreting our results. First, the trajectory data are not guaranteed to be domain-filling, meaning that certain key regions within the WCB may be under-represented or missing entirely. Second, as we have noted, vertical transport by the convection parameterization is not reflected in the grid-scale vertical velocity field in the global simulation, meaning that trajectories do not ascend faster even when they are in a region where the convection parameterisation is triggered. As a result, the convection parameterization may generate high  $N_i$  and low  $q_v$  values in the WCB outflow, while there are no corresponding convective trajectories (although trajectories passing above the convective activity would see the convective tendencies in the moisture variables). In Schwenk and Miltenberger (2025), we therefore verify the findings presented here from a Eulerian perspective and demonstrate that the results remain largely consistent with our finding here.

Code and data availability. Code and data will be made available after publication and will appear under the TPChange community on zenodo: https://zenodo.org/communities/tpchange.


Author contributions. CS and AM designed the experiment and conducted the numerical simulations. AM implemented the online diagnostics, and CS wrote the post-processing code. CS and AM worked jointly on the interpretation of the results. CS drafted the final manuscript with contributions from AM.

Acknowledgements. This work was funded by the Deutsche Forschungsgemeinschaft (DFG, German Research Foundation) through TRR 301 (project no. 428312742; The Tropopause Region in a Changing Atmosphere) sub-project B08 coordinated by Annette Miltenberger. The authors gratefully acknowledge the computing time granted on the supercomputer MOGON 2 at Johannes Gutenberg University Mainz (https://hpc.uni-mainz.de/ last access: 9 December 2024), which is a member of the AHRP (Alliance for High Performance Computing Rhineland-Palatinate, https://www.ahrp.info/, last access: 9 December 2024) and the Gauß-Allianz e.V. We further thank Annika Oertel for useful discussion input and sharing her ICON setup used in Oertel et al. (2023).

## Appendix A: Additional Figures

Figure A1. Longitude and latitude points of trajectories in the beginning (a), end (b) of ascent and at the end of the simulation (c) for the nested (black) and global (red) simulation.

Figure A2. Normalized histograms of  $q_v$  (a) and temperature (b) at the beginning of  $\tau_{WCB}$  for the nested (black) and global (red) simulations.

Figure A3. Trajectory longitude-latitude (a) and latitude-pressure (b) positions at the center of TF1 (00:00 on the 21.09.2017) with color indicating  $\Delta p_{2h}$  for the global simulation.

Figure A4. Trajectory longitude-latitude (a) and latitude-pressure (b) positions at the center of TF2 (18:00 on the 22.09.2017) with color indicating  $\Delta p_{2h}$  for the global simulation.

Figure A5. Trajectory longitude-latitude (a) and latitude-pressure (b) positions at the center of TF3 (18:00 on the 23.09.2017) with color indicating  $\Delta p_{2h}$  for the global simulation.

Figure A6. Plot of all positive  $\Delta p_{2h}$ -values (indicating descent) for each trajectory over simulation run-time for the global (a) and nested (b) simulation.

Figure A7. a) The maximum pressure velocity over  $2 \, \mathrm{h} \, (\mathrm{max}(\Delta p_{2\mathrm{h}}))$  over  $\tau_{600}$ , and b) the vertical velocities experienced by trajectories in TF2 binned by pressure, for the nested simulation in black and the global simulation in red, with mean (solid lines), median (dashed lines), inter-quartile ranges (light shaded) and  $5^{\mathrm{th}}$  to  $95^{\mathrm{th}}$  percentile ranges (dark shaded).

**Figure A8.** Two-dimensional histograms of calculated saturation specific humidity (over ice) given temperature and pressure at the end of the ascent (y axis), over the actual specific humidity (x axis) for the nested (a) and the global (b) simulation.

Figure A9. Behaviour of  $RH_i$  with  $\log_{10}(\tau_{\rm sat,ice})$  for the nested (black) and the global (red) simulation, with mean (solid line), median (dashed line),  $5^{\rm th}$ - $95^{\rm th}$  percentile ranges (light shaded) and inter-quartile ranges (dark shaded). Only trajectories with absolute vertical velocities smaller than  $0.1~{\rm ms}^{-1}$  are included for plotting.

**Figure A10.** Instantaneous microphysical process rates during TF2 binned by pressure (top row) and temperature (bottom row) for riming (a and d), deposition (b and e) and condensation (c and f). Mean (solid line), median (dashed line), inter-quartile ranges (dark shaded) and 5<sup>th</sup> to 95<sup>th</sup> percentile ranges (light shaded) are plotted for the nested (black) and global (red) simulation.

**Figure A11.** Instantaneous microphysical process rates during TF2 binned by temperature for the influx of rain (a), graupel (b), snow (c) and ice (d). Mean (solid line), median (dashed line), inter-quartile ranges (dark shaded) and 5<sup>th</sup> to 95<sup>th</sup> percentile ranges (light shaded) are plotted for the nested (black) and global (red) simulation.

Figure A12. Normalized histograms of pressure (a) and temperature (b) of glaciation for the nested (black) and global (red) simulations.

**Figure A13.** Hydrometeor mass mixing ratios (top row), number concentrations (middle row) and radius (bottom row) for ice (a, d, g), snow (b, e, h) and graupel (c, f, i) during TF2. Mean (solid line), median (dashed line), inter-quartile ranges (dark shaded) and 5<sup>th</sup> to 95<sup>th</sup> percentile ranges (light shaded) are plotted for the nested (black) and global (red) simulation.

**Figure A14.** Hydrometeor mass mixing ratios (top row), number concentrations (middle row) and radius (bottom row) for cloud droplets (a, c, e) and rain during TF2. Mean (solid line), median (dashed line), inter-quartile ranges (dark shaded) and 5<sup>th</sup> to 95<sup>th</sup> percentile ranges (light shaded) are plotted for the nested (black) and global (red) simulation.

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
