# Peer review of "Effects of Model Grid Spacing for Warm Conveyor Belt (WCB) Moisture Transport into the Upper Troposphere and Lower Stratosphere (UTLS)—"

_EGUsphere, 2025_

## Referee Comment (RC1)

**Effects of Model Grid Spacing for Warm Conveyor Belt (WCB) Moisture Transport into the Upper Troposphere and Lower Stratosphere (UTLS) – Part I: Lagrangian Perspective**

by Schwenk and Miltenberger

**General comments**

This paper examines the effect of model grid spacing on the transport of moisture to the upper troposphere and lower troposphere (UTLS) by warm conveyor belts (WCB). To do this, they employ two simulations of a WCB with the ICON model, one at roughly 13 km resolution with parameterized convection and the other at 3.5 km resolution with convection permitted. They compare Lagrangian WCB trajectories and their properties between the two simulations, with links to their Part II Eulerian study (also in preparation). The paper aims to fill the gap in knowledge about how WCB moisture transport changes when transitioning from a convection-parameterizing to a convection-permitting simulation. They find that the convection-permitting simulation produces a drier WCB outflow and illustrate the fact and reasons with a very detailed analysis of trajectories and processes along them.

In general, this is a very well written paper with thorough and detailed analysis presented with nice figures. I do not have much scientific input, but conveying the story could use some adjustments to make it even more convincing.

I have only one major suggestion regarding the structure of the Conclusions and a few minor comments on figures and some phrasing.

**Major Comments**

- The current format of the Conclusions and Discussion section is heavier on the Discussion side and could use an extra paragraph of higher level conclusions. You list the main findings in 1-2 sentences (drier WCB outflow; more pronounced frozen-phase microphysics, stronger frozen precipitation, etc. for the higher resolution simulation) in the Abstract, and I was hoping to see a mirroring paragraph of these high level concluding remarks with some more detail also in the Conclusions and Discussion section. Currently, the rather detailed discussion/conclusion ends at line 600 and continues with placing the study into a larger context and the limitations paragraph. I would recommend adding a paragraph around line 600 that includes the key findings mentioned in the Abstract with some short elaborations. This

would greatly benefit the readers who start from the Conclusions section and would be drawn in to read the full study.

**Minor comments**

- The authors state that the current work is a follow-up study to Schwenk and Miltenberger (2024), pointing out the similarities in the methods and also partly in the results. The presentation of the research question and the hypothesis (the paragraph around line 110) also mentions the earlier study, and I recommend making this the first place where the reader is explicitly informed that the current study is a follow-up study.

- The end of the sentence on lines 128-129 and the beginning of the sentence on line 129 contain the same phrase: 'this study'. I recommend rephrasing these sentences to eliminate any confusion about which study or paper is being referred to.

- The second criterion for the WCB trajectory selection algorithm ("... and second, the trajectories must lie within two specified longitude-latitude regions at two distinct times, determined using Eulerian cloud cover and sea-level pressure data from our simulation.") would benefit from an additional sentence or two that elaborates on how the selection of regions was based on the given Eulerian fields, as this appears to be different from Madonna et al. (2014) and Oertel et al. (2023).

- Figures 2 and 4 might benefit from additional vertical lines that would indicate the mean and/or median values mentioned in the text for the corresponding figures.

- The sentence on line 282 seems to be missing a verb.

- The figure caption for the Figure 5 subplots do not match the actual subplots.

- The subsection 'Hydrometeor Content' in Section 3.2 contains an incorrect figure reference on line 339 and a non-existent figure reference ('Fig. 6g') on line 352.

- Add the word 'trajectories' after the value '0.5%'. The values are very different; however, in the current wording, the 0.5% could refer to either trajectories or the PE value.

**References**

Madonna, E., Wernli, H., Joos, H., and Martius, O.: Warm conveyor belts in the ERA-Interim dataset (1979–2010). Part I: climatology and potential vorticity evolution, J. Clim., 27, 3–26, https://doi.org/10.1175/JCLI-D-12-00720.1, 2014.

Oertel, A., Miltenberger, A. K., Grams, C. M., and Hoose, C.: Interaction of microphysics and dynamics in a warm conveyor belt simulated with the ICOsahedral Nonhydrostatic (ICON) model, Atmospheric Chemistry and Physics, 23, 8553–8581, https://doi.org/10.5194/acp-23-8553-2023, publisher: Copernicus GmbH, 2023.

Schwenk, C. and Miltenberger, A.: The role of ascent timescales for warm conveyor belt (WCB) moisture transport into the upper troposphere and lower stratosphere (UTLS), Atmospheric Chemistry and Physics, 24, 14073–14099, https://doi.org/10.5194/acp-24-14073-2024, 2024.

---

## Referee Comment (RC2)

**Review of "Effects of Model Grid Spacing for Warm Conveyor Belt (WCB) Moisture Transport into the Upper Troposphere and Lower Stratosphere (UTLS) - Part I: Lagrangian"**

by Schwenk and Miltenberger

**General comments:**

In this study, the authors explore the effect of changing horizontal resolution on the representation of warm conveyor belt moisture transport in the ICON model. By employing Lagrangian trajectories, they show that the convection-permitting, finer resolved model simulation results in higher vertical velocities within the WCB. This results in altered cloud microphysical processes and properties compared to the convection-parameterizing simulation.

The manuscript is logically structured and well written. I have some minor technical concerns regarding the setup and the intercomparison of the simulations. Nevertheless, this manuscript merits publication provided that the following comments are addressed.

**Specific comments:**

- P5, L149-151: The choice of two-way coupling introduces a significant challenge for attribution. Because the nested grid feeds back into the global domain, the "nested" setup will naturally diverge from the "global" control, eventually resulting in different synoptic states, in particular towards the end of the simulation period. Consequently, it is intricate to tell whether the reported differences are a direct result of increased resolution or simply a byproduct of this dynamical divergence. I would like the authors to clarify their rationale for this setup, as this could be avoided by employing a one-way coupling where the synoptical state in the global domain is identical between "nested" and "global" setup.
- A potential concern regarding the comparison between the "nested" and "global" simulations is that the trajectories are evaluated at different horizontal resolutions. Without a scale-aware framework, it is difficult to determine if the reported differences are due to resolving finer-scale or merely methodological. To isolate the added value of the finer resolution, I recommend that the authors coarse-grain the nested trajectories by averaging them within the spatial footprint of the global grid boxes. If the discrepancies persist after this upscaling, they can be more confidently attributed to the non-linear effects of resolving finer-scale processes.

**Minor Remarks:**

- P11, L281: *"… at one …";* I assume you mean "… are on …"

- P12, L290-292: Here, the authors state that the underlying distributions are different between the two setups, but you nevertheless report mean values. As the mean is a parametric quantity that is dependent on the underlying distribution, using means to

compare the quantities is only valid if the underlying distributions are equal. I would therefore refer to reporting mean values, but rather median values, as they are independent of the underlying distribution. Please check for further occurrences in the manuscript.

- P15, L362-363: *"… increasing …";* change to "…increases…"

- P19, L472-473: I would also see a second effect that might cause the stronger graupel production. Due to the higher vertical velocities, saturation with respect to water can be more easily sustained, which might to some extent compensate for the depleting effect of the Wegener-Bergeron-Findeisen process for liquid hydrometeors, thereby causing higher graupel production.

- P26, Fig. A1: For better orientation, I would ask the authors to unify the geographical extent and add coastlines.

- P22, L516: *"… below pressures of above 500 hPa …"*; It is not fully clear to me what the authors mean here.